# Viscosity and Structure of a CaO-SiO_2_-FeO-MgO System during a Modified Process from Nickel Slag by CaO

**DOI:** 10.3390/ma12162562

**Published:** 2019-08-12

**Authors:** Yingying Shen, Junkai Chong, Ziniu Huang, Jianke Tian, Wenjuan Zhang, Xingchang Tang, Wanwu Ding, Xueyan Du

**Affiliations:** School of Materials & Science, State Key Laboratory of Advanced Processing and Recycling of Non-ferrous Metals, Lanzhou University of Technology, Lanzhou 730050, China

**Keywords:** nickel slag, basicity, CaO-SiO_2_-FeO-MgO system, viscosity, structure

## Abstract

There is a high iron content in nickel slag that mainly exists in the fayalite phase. Basic oxide can destroy the stable structure of fayalite which is beneficial to the treatment and comprehensive utilization of nickel slag. The research was based on the composition of the raw nickel slag, taking the CaO-SiO_2_-FeO-MgO system as the object and CaO as a modifier. The effect of basicity on the melting characteristics, viscosity and structure of the CaO-SiO_2_-FeO-MgO system was studied. The relationship between the viscosity and structure of the CaO-SiO_2_-FeO-MgO system was also explored. The results show as follows: (1) When the basicity is lower than 0.90, the primary phase of the slag system is olivine phase. When the basicity is greater than 0.90, the primary phase of the slag system transforms into monoxide. When the basicity is 0.90, olivine and monoxide precipitate together as the temperature continues to decrease. At the same time, the liquidus temperature, softening temperature, hemispherical temperature, and flow temperature all reach the lowest value. (2) With the increase of basicity, the critical viscosity temperature of the CaO-SiO_2_-FeO-MgO system decreases first and then increases. Critical viscosity temperature is the lowest at the basicity of 0.90, which is 1295 °C. (3) When the slag system is heterogeneous, the viscosity of the molten slag increases rapidly because of the quantity of solid phase precipitated from the CaO-SiO_2_-FeO-MgO system. (4) When the slag system is in a homogeneous liquid phase, the molar fraction of O^0^ decreases with the increase of basicity and the mole fraction of O^−^, and O^2−^ increases continuously at the basicity of 0.38~1.50. The silicate network structure is gradually depolymerized into simple monomers, resulting in the degree of polymerization, and the viscosity, being reduced. The mole fraction of different kinds of oxygen atoms is converged to a constant value when the basicity is above 1.20.

## 1. Introduction

Iron-rich nickel slag (nickel slag) is the industrial waste discharged from the process of nickel metallurgy in a flash furnace. The production of 1 t of nickel can produce about 6~16 t slag, and the emissions from Jinchuan Group Co. of Gansu Province in China are more than 1.6 million tons per year, with a cumulative stock of up to 40 million tons [1]. The content of TFe (total iron) in nickel slag is up to 40% and valuable metal elements such as Ni, Co, and Cu are also present. Fayalite (2FeO·SiO_2_), which is the main phase in the slag, is a co-melt composed of complex silicates [2]. The structure of fayalite is stable, as it is a complex network crystal in which Si-O atoms are connected with each other, resulting in a low recovery of valuable metal from nickel slag [3]. Adding basic oxides can destroy the network structure of fayalite [4]. If an appropriate amount of CaO is added to the molten nickel slag, the decomposition of the fayalite can be promoted, which is beneficial to the further reduction or oxidation treatment of the nickel slag in order to promote the comprehensive utilization of the nickel slag [1,5,6].

Viscosity is one of the typical physical properties of slag, which not only affects the chemical reaction of slag, transfer rate of elements and precipitation of crystal, but also affects the life of the furnace liner. When the temperature of the slag is lower than the liquidus temperature, the quantity and type of the phase has a significant influence on the viscosity of the slag system; and when the temperature is higher than the liquidus temperature of the slag, the viscosity of the slag system is mainly affected by the slag structure. The viscous performance of the slag is the macroscopic performance of its microstructure. Many scholars have studied the structure and viscosity of different slag systems. Lv et al. [7] studied the viscosity of SiO_2_-MgO-FeO-CaO-Al_2_O_3_ slag system. The results show that when the basicity increases, the degree of slag polymerization and the viscosity both decrease. Talapaneni et al. [8] combined experiments and theory to study the relationship between melt structure and viscosity of high alumina silicates. The results show that the silicate structure in the SiO_2_-MgO-CaO-Al_2_O_3_ slag system was depolymerized with the increase of basicity, resulting in the viscosity and activation energy of the slag being reduced.

More impurities exist in the nickel slag, so the composition of nickel slag is complex. The content of CaO, FeO, SiO_2_ and MgO in nickel slag is as high as 94 wt % [9]. Therefore, the nickel slag system can be simplified to the CaO-SiO_2_-FeO-MgO system according to the composition. Based on the composition of water-quenched nickel slag from enterprise flash furnaces, this paper studied the viscosity and structure of a CaO-SiO_2_-FeO-MgO system during a modified process from nickel slag by CaO, in order to provide a theoretical basis for the modification of nickel slag.

## 2. Experimental Procedure

### 2.1. Experimental Materials

Experimental material: CaO (analytical reagent, Sinopharm Chemical Reagent Co., Ltd., Shanghai, China), SiO_2_ (analytical reagent, Sinopharm Chemical Reagent Co., Ltd.), MgO (analytical reagent, Sinopharm Chemical Reagent Co., Ltd.), FeC_2_O_4_·2H_2_O (analytical reagent, Tianjin Guangfu Fine Chemical Research Institute, Tianjin, China).

Pretreatment of raw materials: (1) Preparation of FeO by FeC_2_O_4_·2H_2_O. FeC_2_O_4_·2H_2_O powder was placed in a high-temperature tube furnace and heated to 1000 °C for 2 h under 300 mL/min of Ar protection. FeC_2_O_4_·2H_2_O tends to break up into FeO, CO, CO_2_ and H_2_O [10,11,12]. After cooling, the powder was ground to 200 mesh or less. (2) SiO_2_, MgO, and CaO were dried in a vacuum drying oven at 1000 °C for 2 h.

### 2.2. Experimental Method

According to the chemical composition of nickel slag, the CaO-SiO_2_-FeO-MgO system was taken as the research object. The ternary basicity (basicity) of the slag system was a variable, and the formula was *R* = (*w*CaO + *w*MgO)/*w*SiO_2_. The content of CaO, MgO, SiO_2_ and FeO in raw nickel slag were 3.77%, 8.86%, 33.31%, and 54.06% respectively, after the conversion of percentage, and the basicity was 0.38. In the experiment, CaO was used as a modifier. With the increase of the CaO content in the modified slag system, the basicity was 0.60, 0.90, 1.20 and 1.50, respectively. The chemical composition of the modified slag system designed is shown in Table 1.

Because the slags were inside the Al_2_O_3_ crucible for a relatively long time for premelting and viscosity measurements, dissolution of Al_2_O_3_ could cause changes in slag composition. And there were many studies that the viscosity of silicate slags varies with the addition of Al_2_O_3_ [13,14,15]. In this paper, the slag composition after viscosity measurements had been tested by ICP (Inductively coupled plasma-atomic emission spectrometry), as shown in the Table 2. It could been seen from the table that the dissolution of Al_2_O_3_ was nothing serious (less than 5%) and the slag composition after viscosity measurements had little change. Thus, the effect of Al_2_O_3_ was negligible in our research.

#### 2.2.1. Preparation of the Pre-Melted Slag Sample

The preparation steps of pre-melting the slag sample were as follows: (1) The pre-treated chemical reagent was weighed according to the data in Table 1, and then put in a mortar for 30 min to ensure the composition was mixed uniformly. (2) The uniform chemical reagent was pressed into a cylinder with the diameter of 30 mm and the height of 10 mm under the pressure of 20 MPa. Then it was placed into a corundum crucible. (3) The crucible containing the sample was put into the high-temperature tube furnace with the temperature raised to 1550 °C at a heating rate of 3 °C/min. After 2 h of heat preservation, it was cooled to room temperature at a cooling rate of 3 °C/min. Throughout the process, 300 mL/min of Ar was used as the protecting gas. (4) The pre-melted slag was crushed to 200 mesh (0.074 mm).

#### 2.2.2. Determination of the Characteristic Temperature

The characteristic temperature was measured using the LZ-III slag melting temperature characteristic tester(Northeast University, Shenyang, China) shown in Figure 1. The experimental method was as follows: (1) Weigh 10g of pre-slag and press the sample with a sample mold. The sample was a cylinder with a diameter of 3 mm and a height of 3 mm. (2) The prepared sample was placed in the LZ-III slag melting temperature characteristic tester. (3) With the temperature increased at a rate of 5 °C/min, the corresponding temperature was recorded as the softening temperature, hemisphere temperature and flow temperature, when the sample dropped to 75%, 50%, and 25%, respectively [16,17]. Throughout the process, 300 mL/min of Ar was used as the protecting gas.

The characteristic temperature points to be observed were as follows: (1) Softening temperature, the temperature at which the sample had fused down to a spherical lump in which the height was 75% of original sample. (2) Hemisphere temperature, the temperature at which the sample had fused down to a hemisphere lump in which the height was 50% of original sample. (3) Flow temperature, the temperature at which the fused mass had spread out in a nearly flat layer with a maximum height of 25% of the original sample. Figure 2 shows the height changes from the slag melting process. These three temperatures characterize the melting trajectory of flux in industrial applications. The hemispherical temperature is referred to as the melting temperature of the mold flux [16,18].

#### 2.2.3. Measurement of the Viscosity

A diagram of the experimental apparatus used to take viscosity measurements is shown in Figure 3. This technique was widely used due to its relative simplicity and reproducibility [19]. The experiments were conducted in a corundum tube using the RTW-16 high temperature melt property tester (Northeast University, Shenyang, China). A B-type thermocouple and a proportional integral differential control system were used to maintain the target temperature. The experimental method was as follows [20]: (1) Weigh 130 g of pre-slag and press into a cylinder with a diameter of 30 mm and a height of 10 mm at a pressure of 20 MPa using a tableting machine. (2) Then place it in a corundum crucible with a diameter of 40 mm and a height of 120 mm. (3) 1.5 L/min of the high purity argon gas was injected into the tube to control the atmospheric conditions. The gases were passed through soda lime to remove excess moisture. (4) Place the crucible into the RTW-16 high temperature melt property tester in the hot zone of the furnace and the temperature of the hot zone was controlled to 1500 °C and was maintained for 1 h to achieve thermal equilibrium of the slag. (5) During the cooling process, the slag viscosity was measured by the rotational torque method. After the spindle was placed in the slag, the viscosity was measured by lowering the temperature at a rate of 5 °C /min. The rotational speed of the spindle was fed back to the computer and the viscosity value was continuous recorded until it reached the range of 2.5~3 Pa·s. It should be noted that, because the pure Ar gas was blown, the low oxygen partial pressure was maintained and most of the Fe would be present as Fe^2+^. The temperature was calibrated before each experiment using a B-type thermocouple inserted from the top of the reaction tube. Castor oil was used as the material for calibration before the viscosity measurements.

#### 2.2.4. Preparation of the Water Quenching Slag Sample

Ten grams of the pre-melted slag was weighed and put into a corundum crucible. Then it was placed into the vertical quenching furnace and heated to 1500 °C at a rate of 3 °C/min. After holding for 2 h, the molten slag was quenched by water. Throughout the process, 300 mL/min of Ar was used as the protecting gas. The water-quenched slag sample was dried in a vacuum oven for 10 h, then crushed and sieved to 200 mesh (0.074 mm). The microstructure was analyzed by FT-IR (Fourier Transform Infrared Spectroscopy) and XPS (X-ray Photoelectron Spectroscopy).

### 2.3. Calculation Method by FactSage.

In this paper, the influence of basicity on the phase diagram of the CaO-SiO_2_-FeO-MgO system was calculated by the module of Phase Diagram in FactSage 7.1 thermodynamic software. The precipitation of the phase in the slag system during non-equilibrium solidification under different basicity was calculated by the module of Equilib.

## 3. Results and Discussion

### 3.1. Effect of Basicity on the Melting Characteristics of the CaO-SiO_2_-FeO-MgO System

Figure 4 shows the phase diagram of the CaO-SiO_2_-FeO-MgO slag system in an argon atmosphere. The totle mass of MgO, FeO and SiO_2_ is marked ‘*Z*’, thus, FeO/*Z* and SiO_2_/*Z* was 0.5618 and 0.3461 in the system. Temperature versus CaO/*Z* is shown in Figure 4. It can be seen that when the basicity is lower than 0.90, the primary phase of the slag system is olivine phase. When the basicity is greater than 0.90, the primary phase of the slag system transforms into monoxide. With the basicity increased, the area of the olivine phase gradually decreases, and the liquidus temperature decreases first and then increases. When the basicity is 0.90, the liquidus temperature is about 1297 °C and reaches the lowest temperature, olivine and monoxide precipitate together as the temperature continues to decrease.

Figure 5 shows the effect of basicity on the characteristic temperature of a CaO-SiO_2_-FeO-MgO system. It can be seen from the figure that when the basicity is 0.38~1.50, the softening temperature, hemisphere temperature and flow temperature of the slag system decrease first and then increase with the increase of basicity. When the basicity is 0.90, the softening temperature, hemisphere temperature and flow temperature are all the lowest, which are 1244, 1256 and 1274 °C respectively. The reason is analyzed as follows. When the basicity is 0.38~0.90, with the increase of basicity, CaO destroys the network structure of the olivine phase, resulting in the olivine phase gradually disintegrating [21,22]. The effect of CaO is conducive to the lowering of the liquidus temperature of the slag system, as the softening temperature, hemisphere temperature and flow temperature are also reduced. When the basicity is 0.90~1.50, CaO itself has a high melting point and is easy to combine with other components in the slag system to form the high melting point substance, so that the liquidus temperature, softening temperature, hemispherical temperature and flow temperature of the system are gradually increased. The flow temperature is significantly higher than the softening temperature and the hemispherical temperature, resulting in the deterioration of fluidity.

### 3.2. Effect of Basicity on the Viscosity of a CaO-SiO_2_-FeO-MgO System

#### 3.2.1. Effect of Basicity on the Viscosity-Temperature Curve of a CaO-SiO_2_-FeO-MgO System

Figure 6 shows the viscosity-temperature curve of a CaO-SiO_2_-FeO-MgO system at different basicities. It can be seen from the figure that the viscosity of the slag system is all lower than 0.25 Pa·s when the temperature is higher than 1400 °C, with the basicity in the range of 0.38~1.50. At the same basicity condition, the temperature decreases, and the viscosity change of the slag system is not significant. The reason is analyzed as follows. When the basicity is 0.90, the liquidus temperature of the modified slag system is the lowest, which is 1297 °C. When the basicity is 1.50, the liquidus temperature of the modified slag system is the highest, which is 1394 °C. When the temperature is above the liquidus temperature, the slag is a homogeneous system. And the temperature is below the liquidus temperature, the slag is a heterogeneous system [23]. When the temperature is higher than 1400 °C, the slag system is in a uniform liquid phase at the range of 0.38~1.50. The microstructure of the slag plays a leading role in the viscosity of the slag system. The effect of the slag microstructure on the slag viscosity is weak, resulting in less change in viscosity. When the temperature is lower than 1400 °C, the system is a heterogeneous slag system. With the decrease in temperature, the viscosity of the slag system with different basicities at different temperatures is mainly related to the type and quantity of the precipitate phase from the slag system. At the same time, when the viscosity of the slag is higher than 0.25 Pa·s, it increases rapidly with the decrease of temperature, which has the characteristic of typical crystallization slag at the basicity of 0.38, 0.60, 0.90, and 1.20. When the viscosity is higher than 0.75 Pa·s, viscosity of the slag begins to rise sharply with the temperature decrease and has typical plastic slag characteristics at the basicity of 1.50, because of the high basicity.

Figure 7 shows the effect of basicity on the viscosity of the CaO-SiO_2_-FeO-MgO system at different temperatures. It can be seen from the figure that the viscosity of the slag system gradually decreases with the increase of basicity when the slags are all above the liquidus temperature (1400~1500 °C). At the same basicity, the viscosity of the slag is gradually reduced with the increase of temperature. At 1500 °C, when the basicity increases from 0.38 to 0.60, the viscosity decreases rapidly from 0.14 Pa·s to 0.11 Pa·s. When the basicity increases from 0.60 to 1.50, the viscosity decreases slowly from 0.11 Pa·s to 0.08 Pa·s. With the increase of basicity, the trend of viscosity reduction is consistent when the temperature is in the range of 1400~1500 °C.

#### 3.2.2. Effect of Basicity on Critical Viscosity Temperature of a CaO-SiO_2_-FeO-MgO System

As the slag is gradually cooled from the liquid phase, the viscosity gradually increases. When the temperature drops to a certain point, the viscosity will increase rapidly, this temperature point is called the critical viscosity temperature (*T*_cv_) [24]. It is well known that viscosity measurements taken at temperatures higher than the critical viscosity temperature will be in the fully liquid region of the system. To highlight the critical viscosity temperature and the experimental test region, the viscosity as a function of temperature in the CaO-SiO_2_-FeO-MgO system at the basicity of 0.38 is given as an example in Figure 8. The critical viscosity temperature is known as the temperature at which the slope changes in the Arrhenius plot (ln(viscosity)-1/*T* graph). The natural logarithm of viscosity as a function of reciprocal temperature shows a significant increase below temperatures of 1380 °C (1653 K), which is the solid-liquid coexisting region, and it suggests the formation of solid precipitates [25]. Thus, the experimental measurements were taken above the critical viscosity temperature as described in Figure 8. In the same way, at the basicity of 0.60, 0.90, 1.20, and 1.50, critical viscosity temperatures are 1360, 1295, 1372 and 1393 °C, respectively.

Figure 9 shows the effect of basicity on the critical viscosity temperature of a CaO-SiO_2_-FeO-MgO system. It can be seen that critical viscosity temperature of the slag system decreases firstly and then increases with the increase of basicity at the range of 0.38~1.50. And critical viscosity temperature is the lowest, which is 1295 °C at the basicity of 0.90. In order to ensure the migration of components in the slag and strengthen the dynamic conditions of the reaction, a lower viscosity and good fluidity of the slag system at the smelting temperature range is beneficial to the process of metallurgical production. So it is necessary to reduce critical viscosity temperature of the slag system.

#### 3.2.3. Effect of Phase Precipitation on the Viscosity of the CaO-SiO_2_-FeO-MgO System

Figure 10 is the XRD (X-ray Diffractometry) patterns of the CaO-SiO_2_-FeO-MgO system by water quenching at 1290 °C. It can be seen that the primary crystal phase precipitated from the slag system is FeO (monoxide) when the basicity is 1.50, and (Mg,Fe)_2_SiO_4_ (olivine) when the basicity is 0.38. At the basicity of 0.90, FeO and (Mg,Fe)_2_SiO_4_ exist simultaneously. The experimental result is in agreement with the phase diagram analysis results.

Figure 11 shows the effect of solid precipitation in the modified slag system on viscosity of the slag. Because the viscosity of the slag is affected by the contents of solid phase and liquid phase at high temperature, and the former is more significant [26]. When the slag system continues to cool below the liquidus temperature, the solid phase can be precipitated because of the saturation as the temperature decreases and continue to be formed. At this time, the viscosity of the slag will increase rapidly. The formation rate of the solid phase is smaller when the temperature is higher than the critical viscosity temperature of the slag system. By contrast, the formation rate of the solid phase is significantly increased when the temperature is lower than critical viscosity temperature.

### 3.3. Effect of Basicity on the Structure of the CaO-SiO_2_-FeO-MgO System

Figure 12 shows the schematic disintegration of the silicate structure by basic oxides. There are three types of oxygen in the molten slag, shown as follows [27]. (1) Bridging oxygen (O^0^, the oxygen atom is connected to two silicon atom). (2) Non-bridging oxygen (O^−^, the oxygen atom is connected to a silicon atom). (3) Free oxygen (O^2−^, the oxygen atom is not connected to the silicon atom). With the addition of basic oxides, free oxygen will enter the silicate system and destroy the bridge oxygen bonding in the silicate structure. Thereby the complicated silicate network structure becomes a simple silicate structure and a large number of non-bridged oxygen forms. For example, the main structure of [Si_3_O_9_]^6−^ combines with the free oxygen released by the basic oxide to decompose into a dimeric structure of [Si_2_O_7_]^6−^ and a monomeric structure unit of [SiO_4_]^4−^, while the cation of the basic oxide will act as a balancing charge. Q^3^, Q^2^, Q^1^, and Q^0^ indicate that the number of bridge oxygens connected to each silicon atom is 3, 2, 1, and 0.

Figure 13 shows the FT-IR spectra of water quenching slag sample at 1500 °C. The [SiO_4_]^4−^ tetrahedron symmetric stretching vibration zone is in the range of 800~1200 cm^−1^, the four kinds of Si-O zones are in 1100~1150 cm^−1^ (Q^3^, layered), 950~980 cm^−1^ (Q^2^, chain), 900~920 cm^−1^ (Q^1^, dimer), and 850~880 cm^−1^ (Q^0^, monomer) [28,29]. It can be seen from the figure that as the basicity increases from 0.38 to 1.50, the center position of the [SiO_4_]^4−^ tetrahedral symmetric stretching vibration zone gradually shifts to the low wavenumber region and the valley depth of the [Si-O] bending vibration zone is gradually reduced. At the same time, the valley depth corresponding to the central position of Q^0^ is gradually deepened and the valley depth corresponding to the central position of Q^3^ is gradually shallower, but the valley depth corresponding to the central position of Q^1^ and Q^2^ is not obvious. It indicates that CaO as a network modification gradually destroys the complex network structure of silicate into a simple dimer (Q^1^) or monomer (Q^0^), which leads to the continuous decrease of the polymerization degree of silicate.

The peak of O_1s_ in the XPS spectrum can be decomposed into the peaks corresponding to three different kinds of oxygen atoms. For the different basicites from 0.38 to 1.50, the accuracies (r^2^) of deconvolution of the XPS peak are 21.78, 22.13, 23.32, 24.97 and 27.73, respectively. The XPS peak was calibrated through Au calibration [30,31]. The envelope of the O_1s_ peak of the water quenching slag samples at different basicities are divided by the Gaussian function. The distribution information of the characteristic peaks in this region is shown in Figure 14. The ratio of the integral area of the peak corresponding to different kinds of oxygen atoms to the integral area of the O_1s_ peak is the mole fraction. The positions and mole fractions of O^0^, O^−^, O^2−^ at different basicities are shown in Table 3.

Figure 15 shows the effect of basicity on the mole fraction of different oxygen atoms in CaO-SiO_2_-FeO-MgO water-quenched slag samples. It can be seen that when the range of the basicity is 0.38~1.50, the molar fraction of O^0^ decreases with the increase of basicity, and the mole fraction of O^−^ and O^2−^ increases continuously. The molar fractions of O^0^, O^−^ and O^2−^ changed significantly at the basicity of 0.38~1.20. But the molar fractions of O^0^, O^−^ and O^2−^ are almost unchanged when the basicity continues to increase to 1.50. It is shown that with the increase of basicity, the silicate network structure is gradually depolymerized into simple monomers, resulting in the degree of polymerization being reduced, and the viscosity gradually reduced, which is consistent with the results of the FT-IR spectra analysis. At the same time, it also explains the reason for the viscosity decreasing gradually with the increased basicity in Figure 7 from the level of microstructure.

Equation (1) is a fundamental result of the charge balance required by the tetrahedral co-ordination of oxygen with silicon, and when any silicate anions associate to form higher polymers plus oxygen ions, the overall reaction reduces to Equation (1).

2O^−^ → O^0^ + O^2−^(1)

Under equilibrium conditions, an equilibrium constant k may be written according to Equation (2) [32]. The curve of k with basicity of the CaO-SiO_2_-FeO-MgO system is as shown in Figure 16. It can be seen from the figure that the mole fraction of different kinds of oxygen atom converges to a constant value when the basicity is above 1.20.

(2)$$k=\frac{\left(O^{0})\;(O^{2-}\right)}{{\left(O^{-}\right)}^{2}}$$

## 4. Conclusions

The viscosity and structure of the CaO-SiO_2_-FeO-MgO system during a modified process from nickel slag by CaO at the basicity of 0.38~1.50 is systematically studied in this paper. The results show as follows.
(1)When the basicity is lower than 0.90, the primary phase of the slag system is olivine phase. When the basicity is greater than 0.90, the primary phase of the slag system transforms into monoxide. When the basicity is 0.90, olivine and monoxide precipitate together as the temperature continues to decrease. At the same time, the liquidus temperature, softening temperature, hemispherical temperature, and flow temperature all reach the lowest value.(2)With the increase of basicity, critical viscosity temperature of the CaO-SiO_2_-FeO-MgO system decreases first and then increases. Critical viscosity temperature is the lowest at the basicity of 0.90, which is 1295 °C.(3)When the slag system is heterogeneous, the viscosity of the molten slag increases rapidity because of the quantity of solid phase precipitated from the CaO-SiO_2_-FeO-MgO system.(4)When the slag system is in a homogeneous liquid phase, the molar fraction of O^0^ decreases with the increase of basicity and the mole fraction of O^−^and O^2−^ increases continuously at the basicity of 0.38~1.50. The silicate network structure is gradually depolymerized into simple monomers, resulting in the degree of polymerization being reduced and the viscosity being reduced as well. The mole fraction of different kinds of oxygen atoms is converged to a constant value when the basicity is above 1.20.

## Figures and Tables

**Figure 1 materials-12-02562-f001:**
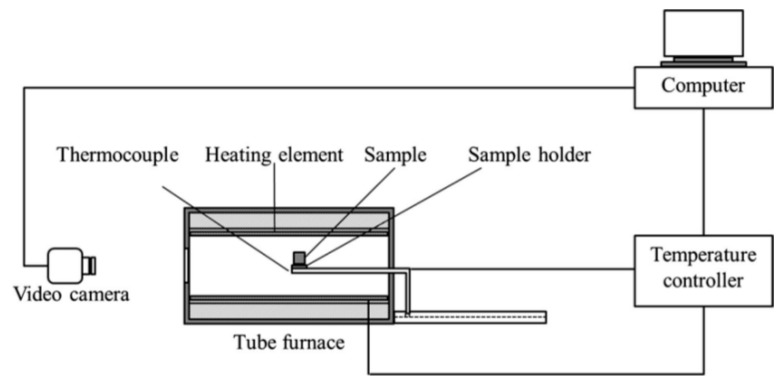
Diagram of the LZ-III slag melting temperature characteristic tester.

**Figure 2 materials-12-02562-f002:**
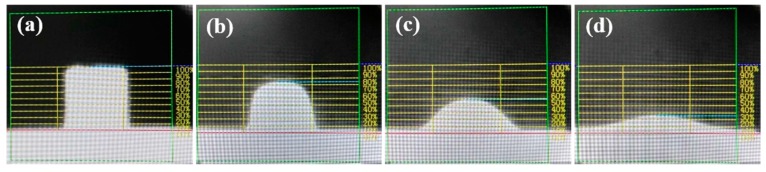
The height changes of slag melting process. (**a**) The original height; (**b**) The corresponding height of softening temperature; (**c**) The corresponding height of hemispheric temperature; (**d**) The corresponding height of flow temperature.

**Figure 3 materials-12-02562-f003:**
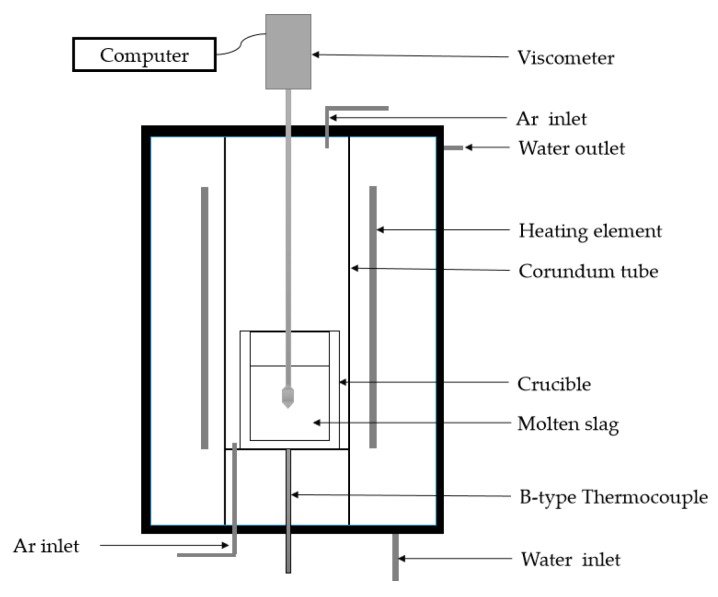
Diagram of the experimental apparatus used for viscosity measurements.

**Figure 4 materials-12-02562-f004:**
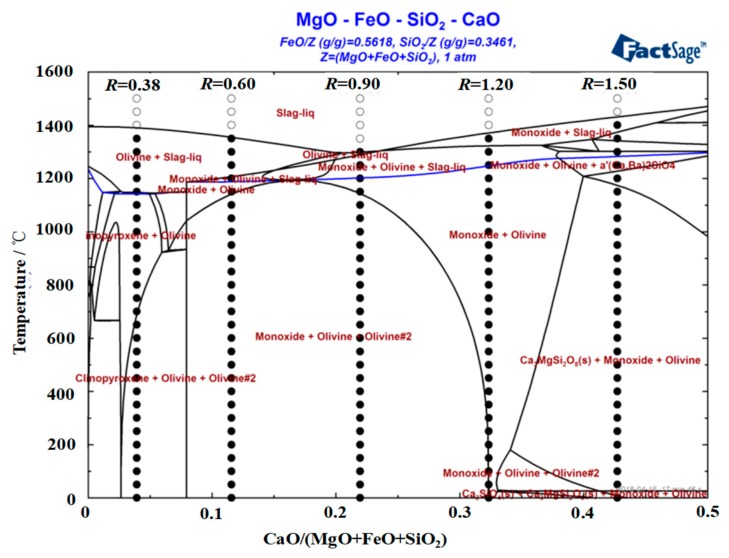
Phase diagram of a CaO-SiO_2_-FeO-MgO slag system in an argon atmosphere.

**Figure 5 materials-12-02562-f005:**
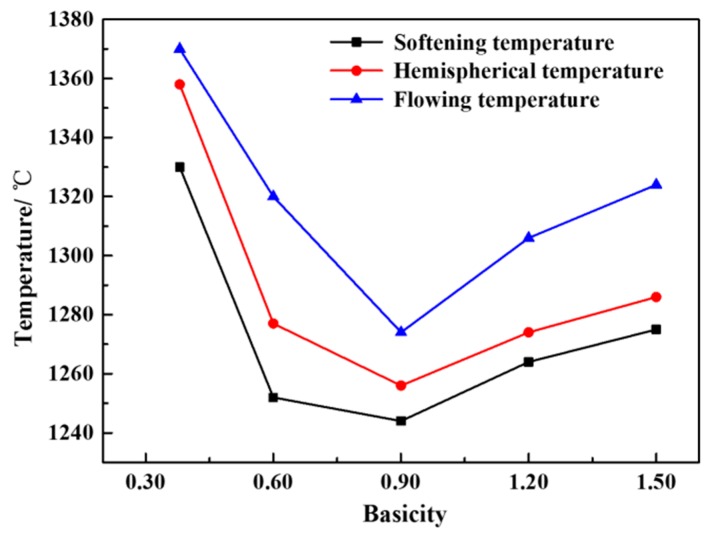
Effect of basicity on characteristic temperature of a CaO-SiO_2_-FeO-MgO system.

**Figure 6 materials-12-02562-f006:**
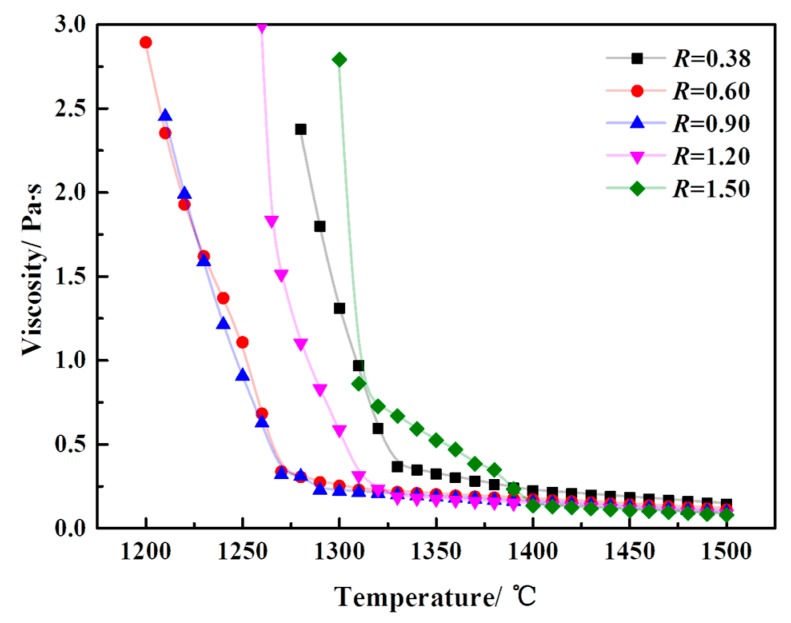
The viscosity-temperature curve of a CaO-SiO_2_-FeO-MgO system at different basicities.

**Figure 7 materials-12-02562-f007:**
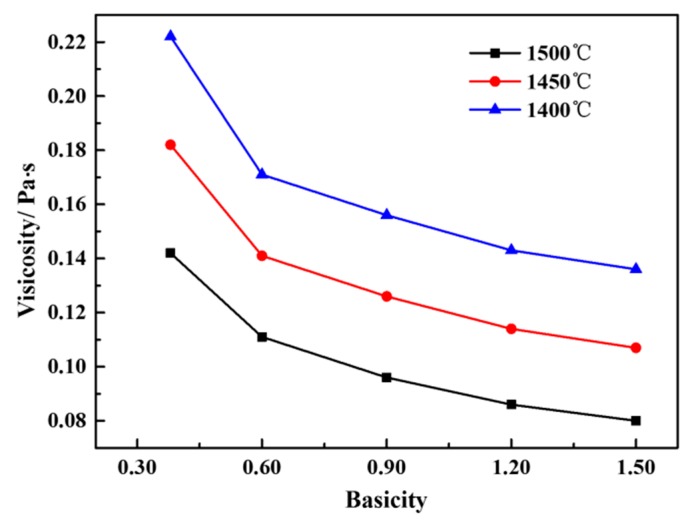
The curve between basicity and viscosity of slag at different temperatures.

**Figure 8 materials-12-02562-f008:**
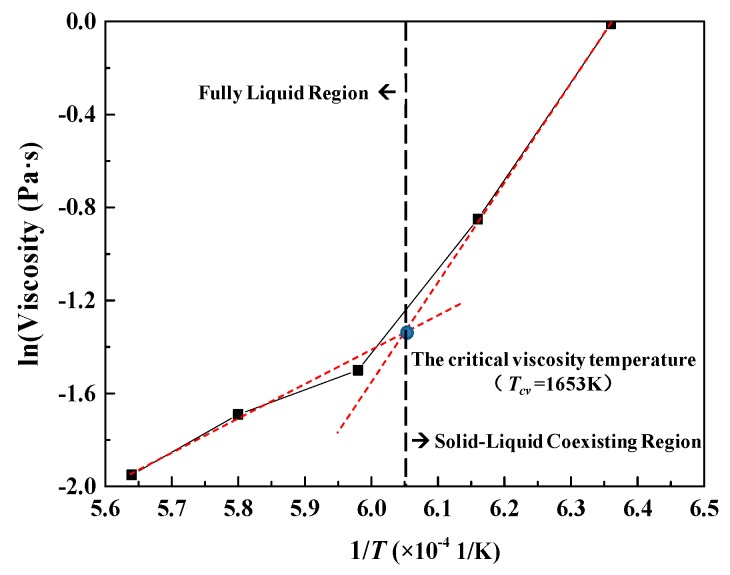
Example of the critical viscosity temperature for the CaO-SiO_2_- FeO-MgO system at the basicity of 0.38. Note the critical viscosity temperature is 1380 °C (1653 K).

**Figure 9 materials-12-02562-f009:**
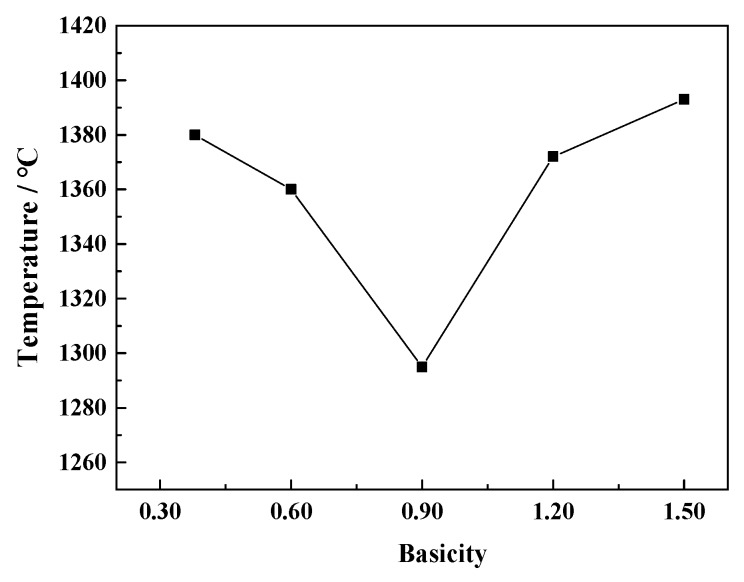
Relationship between basicity and critical viscosity temperature of the CaO-SiO_2_-FeO-MgO system.

**Figure 10 materials-12-02562-f010:**
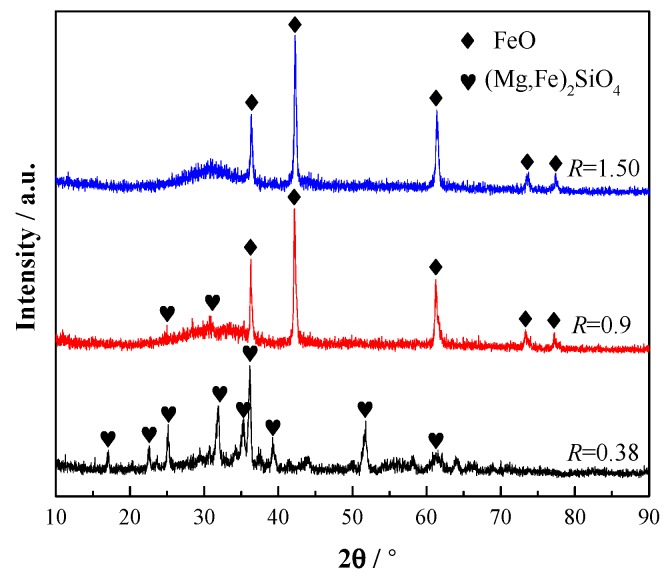
XRD spectrum of the CaO-SiO_2_-FeO-MgO system.

**Figure 11 materials-12-02562-f011:**
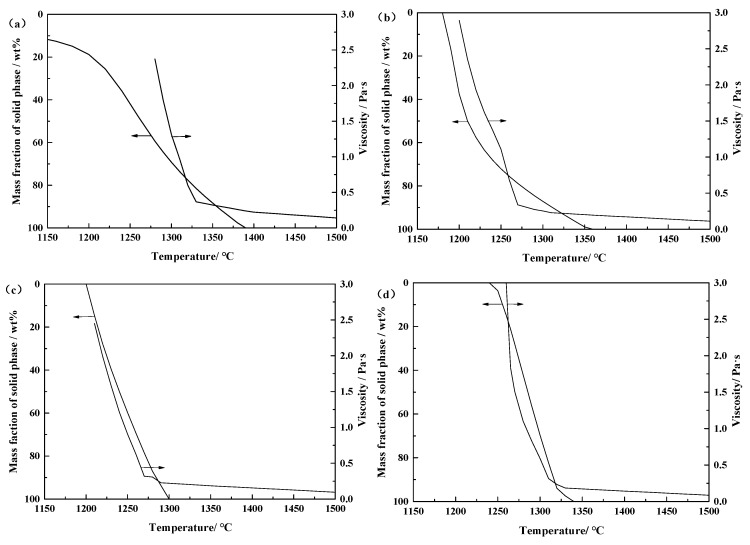
Effect of solid phase precipitation on the viscosity of the CaO-SiO_2_-FeO-MgO system at different basicities: (**a**) *R* = 0.38; (**b**) *R* = 0.60; (**c**) *R* = 0.90; (**d**) *R* = 1.20; (**e**) *R* = 1.50.

**Figure 12 materials-12-02562-f012:**
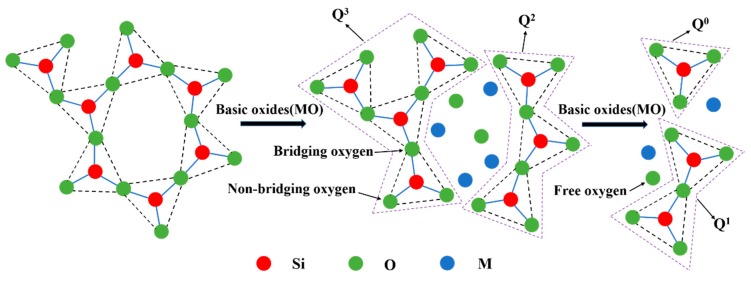
Schematic disintegration of the silicate structure by basic oxides.

**Figure 13 materials-12-02562-f013:**
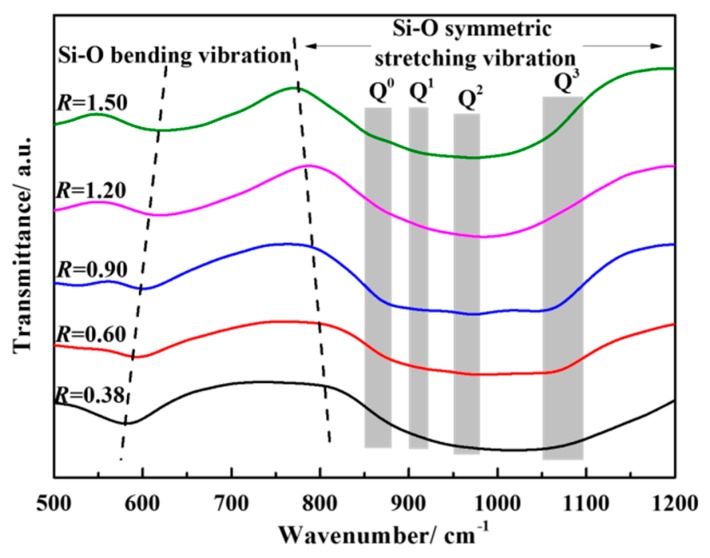
FT-IR spectra of the water quenching slag sample.

**Figure 14 materials-12-02562-f014:**
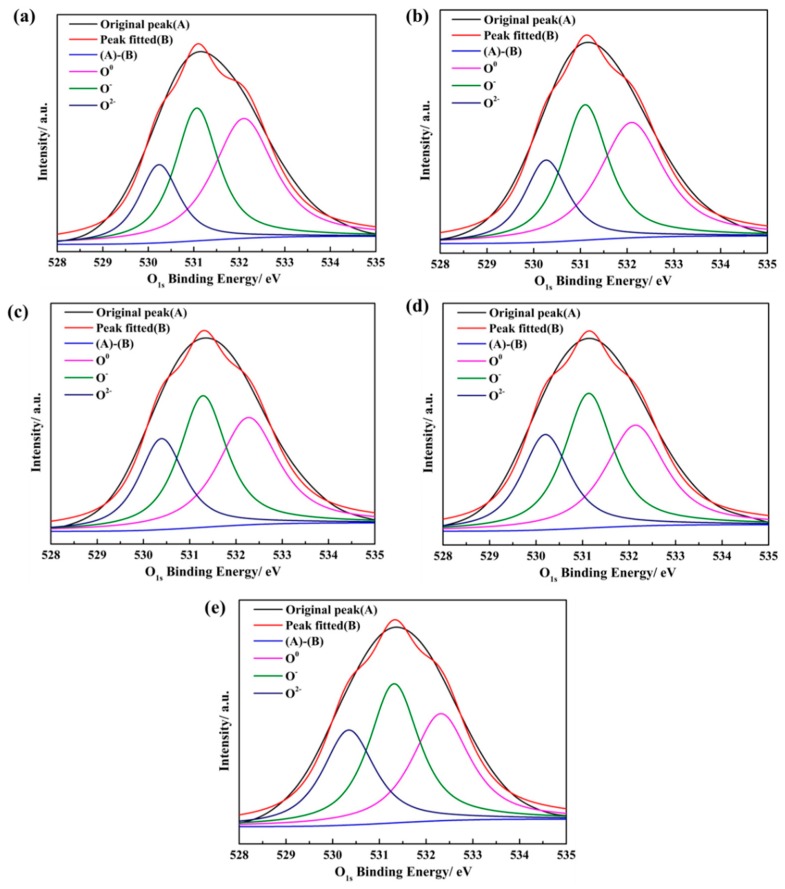
The XPS peak fitting spectrum of O_1s_ in water quenching slag sample at different basicities: (**a**) *R* = 0.38; (**b**) *R* = 0.60; (**c**) *R* = 0.90; (**d**) *R* = 1.20; (**e**) *R* = 1.50.

**Figure 15 materials-12-02562-f015:**
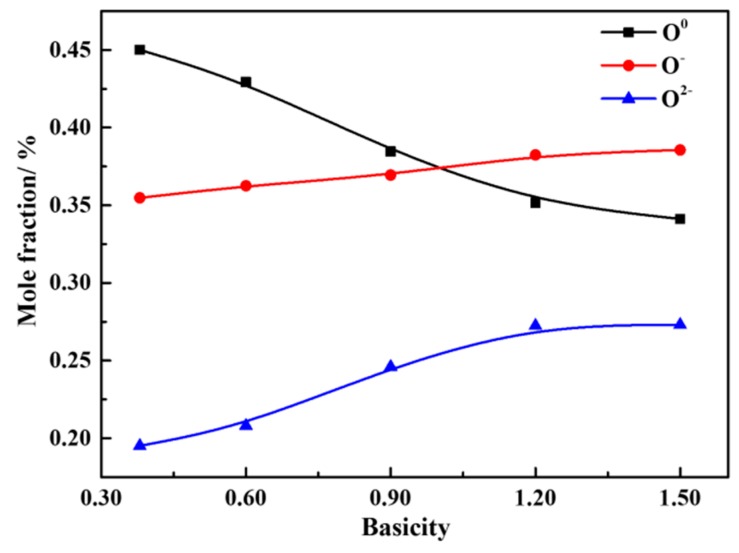
Curve of the mole fraction of different kinds of oxygen atoms with the basicity.

**Figure 16 materials-12-02562-f016:**
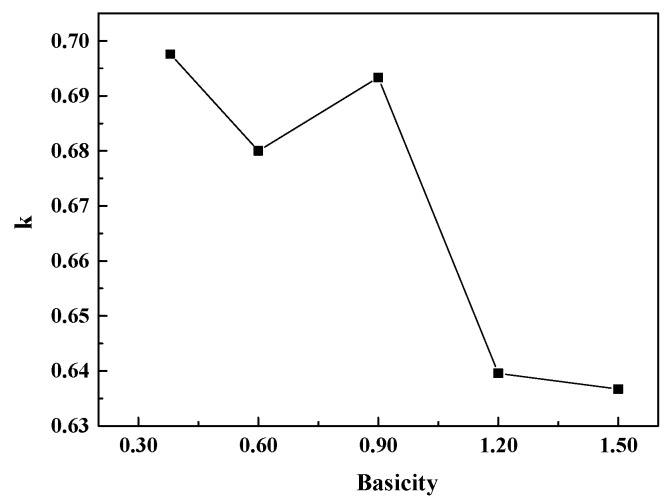
The curve of k with basicity of the CaO-SiO_2_-FeO-MgO system.

**Table 1 materials-12-02562-t001:** Chemical composition of modified slag system (wt %).

Slag System	CaO	MgO	SiO_2_	FeO	Basicity
S1	3.77	8.86	33.31	54.06	0.38
S2	10.36	8.26	31.03	50.35	0.60
S3	17.99	7.75	28.39	46.07	0.90
S4	24.43	6.96	26.15	42.46	1.20
S5	29.93	6.45	24.25	39.37	1.50

**Table 2 materials-12-02562-t002:** The slag composition after viscosity measurements (wt %).

Slag System	CaO	MgO	SiO_2_	FeO	Al_2_O_3_
S1	3.72	8.68	32.78	51.54	3.28
S2	10.28	8.14	30.01	47.51	4.06
S3	16.72	7.65	28.10	43.40	4.13
S4	22.41	6.82	25.11	41.48	4.18
S5	29.84	6.16	24.17	35.31	4.52

**Table 3 materials-12-02562-t003:** Fitting positions and mole fractions of different types of oxygen atoms.

Basicity	O^0^		O^−^		O^2−^	
	Position (eV)	Mole Fraction	Position (eV)	Mole Fraction	Position (eV)	Mole Fraction
*R* = 0.38	532.10	45.01%	531.07	35.48%	530.24	19.51%
*R* = 0.60	532.09	42.94%	531.10	36.25%	530.27	20.81%
*R* = 0.90	532.27	38.46%	531.29	36.94%	530.39	24.60%
*R* = 1.20	532.14	35.16%	531.13	38.24%	530.20	26.60%
*R* = 1.50	532.31	34.12%	531.31	38.56%	530.34	27.31%

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
