# Peer review of "Viscosity and Structure of a CaO-SiO2-FeO-MgO System during a Modified Process from Nickel Slag by CaO"

_materials, 2019, doi:10.3390/ma12162562_

Round 1

Reviewer 1 Report

This study identifies the physical properties such as melting and viscous behavior and ionic structure of nickel slag, which helps to understand the phenomena occurring in modified process by CaO. However, there are some deficiencies in the considerations listed below, which need to be corrected and confirmed. Mandatory revisions are suggested.

1)      The authors conducted discussions on the relationship between ionic structure and physical properties of nickel slags via melting of the oxides. In experimental procedure (page 2 line 64), it would be better to indicate the manufacturer of the oxide. In addition, it is needed to explain why FeC2O4 was used to preparation of FeO. Did the authors expect the reduction of FeO by carbon?

2)      The authors placed the slags inside the Al2O3 crucible for a relatively long time for pre-melting and viscosity measurements (page 3 line 84). Dissolution of Al2O3 may cause changes in slag composition. It would be better to indicate the XRF results of the final slag composition.
Please check the following references.
(Park, J.M.; Keum, C.H.; Son, J.W.; Shin, Y.K. Reaction equilibria between molten iron and CaO-MgOsatd.-FetO-SiO2-MnO(2-30wt%)-MxOy slag[J], Steelmaking conference proceedings, 1994, 77: 461-470)

3)      The authors investigated the melting temperature by "Slag Melting Temperature Characteristic Tester.” It would be better to cite the appropriate references to explain the experimental method. Determine the softening temperature, hemisphere temperature, and flow temperature for readers.
Please check the following references.
(ASTM D 1857-68, Standard Test Method for Fusibility of Coal and Coke Ash,1985 Annual Book of Standards, 1985, 5: 298-)

4)      The authors performed viscosity measurements using the rotational torque method (page 3 line 95). It would be better to indicate the detailed method for readers. For example, the calibration method and the stepped-temperature-viscometry technique are needed to explain.
Please check the following references.
(Lee, S; Min, D.J. Viscous Behavior of FeO‐Bearing Slag Melts Considering Structure of Slag[J]. Steel Research International, 2018, 89(8): 1800055.)

5)      The authors calculated the phase diagram using Factsage 7.1 (page 4 line 118). Was the oxygen partial pressure considered? (FeO=Fe+1/2O2) In phase diagram module of Factsage 7.1, the oxygen partial pressure can be fixed and calculated. Also, drawing the five phase diagrams (Figure 2 a-e) as binary phase diagram seems to be clearly expressed. For example, it would be better for the reader to express binary phase diagram (Temperature versus CaO/SiO2) with fixed MgO/SiO2 and FeO/SiO2.

6)      Please explain the relationship between viscosity and melting temperature around the olivine-monoxide eutectic point (page 5 line 136). How does the ionic structure of slag affect the increase of melting point?

7)      The authors derive the critical temperature(TC) from the temperature-viscosity graph. The critical temperature is known as the temperature at which the slope changes in the Arrhenius plot (ln(viscosity)-1/T graph). In particular, the tangent line in Figure 6 (e) is controversial. In addition, for a discussion of the relationship between solid fraction and viscosity (Figure 8), a detailed discussion of the relationship between Einstein-roscoe equation and critical temperature is needed.
Please check the following references.
(Park, H.S.; Park, S.S.; Sohn, I. The viscous behavior of FeOt-Al2O3-SiO2 copper smelting slags[J], Metallurgical and Materials Transactions B, 2011, 42: 692-699)

8)      Why did not the critical temperature (Figure 7) show together with Figure 3? Also, why are the trends different (B=0.6)?

9)      The authors stated relationship between the formation rate of solid phase and the critical temperature. (page 8 line 210) However, the formation rate is kinetic term. Please discuss more correlation kinetic and thermodynamic property of slags.

10)  The authors explained that the solid fraction of slags using Factsage calculation(page 9 line 226). To demonstrate the phase-property relationship, it would be better to indicate the experimental evidence such as XRD analysis. To apply the roscoe-equation, it would be better to calculate the total solid fraction of Figure 8.

11)   The authors investigate the XPS analysis to comprehend the ionic structure of slags (page 11 line 256). What is the accuracy(r2) of deconvolution of XPS peak? The original peak and the peak fitted appear to be different. It is also needed to explain that the XPS peak was calibrated (C2P or Au calibration).
Please check the following references.
(Park, J.-H.; Rhee, P.C.-H. Ionic properties of oxygen in slag[J],
Journal of Non-crystalline Solids, 2001, 282: 7-14)
(Yamashita, Y.; Hayes, P. Analysis of XPS spectra of Fe2+ and Fe3+ ions in oxide materials[J], Applied Surface Science, 2008, 254: 2441-2449)

12)  Please discuss that the mole fraction of different kinds of oxygen atom was converged to a constant value when basicity is above 1.2. It is necessary to calculate the equilibrium constant value between oxygen atoms(k=(O-)2/(O0)*(O2-)).
Please check the following references.
(Toop, G.W.; Samis, C.S. Activities of ions in silicate melts[J],
Transactions of the Metallurgical Society of AIME, 1962, 224(6): 878-887)

Author Response

Dear reviewer:

We would like to express our gratitude to you for giving us an opportunity to revise our manuscript. We also appreciate you very much for your positive and constructive comments and suggestions on our manuscript entitled “Viscosity and Structure of CaO-SiO2-FeO-MgO System during modified Process from Nickel Slag by CaO”. (Article reference: materials-553686).

We have revised the original manuscript according to the reviewer’s advice, with the corrected positions labeled with highlight, as shown in the corrected version. In addition, a correction list including answers to the referee and corresponding corrections has been attached.

Now we ask you to consider the revised manuscript again, and are looking forward to hearing the good news.

Looking forward to hearing from you.
Thank you and best regards.

Yours sincerely,
Yingying Shen
E-mail: shenyingying005@163.com

Reviewer 2 Report

The viscosity and structure of CaO-SiO2-FeO-MgO system during modified process from nickel slag by CaO at the basicity of 0.38~1.50 is systematically studied in this paper. The results show as follows.
When the basicity is 0.38~0.90, the phase of primary crystal is olivine at argon atmosphere,  while it is monoxide at the basicity of 0.90~1.50.
With the increase of basicity, critical viscosity temperature of CaO-SiO2-FeO-MgO system decreases first and then increases.
When the slag system is heterogeneous, the viscosity of the molten slag increase rapidity because of the type and quantity of solid phase precipitated from CaO-SiO2-FeO-MgO system.
When the slag system is in a homogeneous liquid phase, the molar fraction of O0 decreases with the increase of basicity and the mole fraction of O- and O2- increases continuously at the basicity of 0.38~1.50.

Must correct the following errors:

Line 51. Replace comma by period before "The".

Line 54. Replace comma by period before "The".

The font size in figure 2 is very small and it is not read. Increase size of Figure 2 and the font size and legend in Figure 2.

Line 116. Remove white space between "atmosphere" and "."

Line 210. Replace capital letters with lowercase after "By contrast, the..."

Author Response

(The authors gave the same response as above.)

Round 2

Reviewer 1 Report

This study identifies the physical properties such as melting and viscous behavior and ionic structure of nickel slag, which helps to understand the phenomena occurring in modified process by CaO. With a good revision, the physical properties of nickel slag will be acceptable and worthy for metallurgical process. As a reader, some discussions suggest below.

The authors stated that the dissolution of Al2O3 was nothing serious and the effect of Al2O3 was negligible in this study. However, there are many studies that the viscosity of silicate slags varies with the addition of Al2O3. It would be better to show the final composition.
Please check the following references.
(Zhang, G.-H.; Chou, K.-C. Influence of Al2O3/SiO2 ratio on viscosities of CaO-Al2O3-SiO2 melt[J], ISIJ International, 2013, 53: 177-180)
(Park, J.H.; Kim, H.; Min, D.J. Novel approach to link between viscosity and structure of silicate melts via Darken’s excess stability function: Focus on the amphoteric behavior of alumina[J], Metallurgical and Materials Transactions B, 2008, 39: 150-153)
(Li, J.L.; Shu, Q.F.; Chou, K. Effect of Al2O3/SiO2 mass ratio on viscosity of CaO-Al2O3-SiO2-CaF2 slag[J], Ironmaking & Steelmaking, 2015, 42: 154-160) The authors described the viscosity measurement technique with Figure 4. However, Figure 4 is exactly the same as the Figure in Ref 16, is there no plagiarism problem? The crucible and the spindle in Figure 4 appear to be metal. Also, the viscosity values in Figure 6 appear to be measured at intervals less than 50K(). On Page 4, line 124: please check the phrase “in a corindon reaction tube”.

Author Response

Dear Reviewer: We would like to express our gratitude to you for giving us an opportunity to revise our manuscript again. We also appreciate you very much for your positive and constructive comments and suggestions on our manuscript entitled “Viscosity and Structure of CaO-SiO2-FeO-MgO System during modified Process from Nickel Slag by CaO”. (Article reference: materials-553686). We have revised the original manuscript according to the reviewer’s advice, with the corrected positions labeled with highlight, as shown in the revised manuscript. And the language and grammar have been improved with the help of a native English speaking colleague. In addition, a correction list including answers to the referee and corresponding corrections has been attached. Now we ask you to consider the revised manuscript again, and are looking forward to hearing the good news. Looking forward to hearing from you. Thank you and best regards. Yours sincerely, Yingying Shen E-mail: shenyingying005@163.com
